

# The use of nutritional supplements to induce ketosis and reduce symptoms associated with keto-induction: a narrative review

Cliff J. d C. Harvey, Grant M. Schofield and Micalla Williden

Human Potential Centre, Auckland University of Technology, Auckland, New Zealand

## ABSTRACT

**Background**. Adaptation to a ketogenic diet (keto-induction) can cause unpleasant symptoms, and this can reduce tolerability of the diet. Several methods have been suggested as useful for encouraging entry into nutritional ketosis (NK) and reducing symptoms of keto-induction. This paper reviews the scientific literature on the effects of these methods on time-to-NK and on symptoms during the keto-induction phase.
**Methods**. PubMed, Science Direct, CINAHL, MEDLINE, Alt Health Watch, Food Science Source and EBSCO Psychology and Behavioural Sciences Collection electronic databases were searched online. Various purported ketogenic supplements were searched along with the terms "ketogenic diet", "ketogenic", "ketosis" and ketonaemia (/ ketonemia). Additionally, author names and reference lists were used for further search of the selected papers for related references.
**Results**. Evidence, from one mouse study, suggests that leucine doesn't significantly increase beta-hydroxybutyrate (BOHB) but the addition of leucine to a ketogenic diet in humans, while increasing the protein-to-fat ratio of the diet, doesn't reduce ketosis. Animal studies indicate that the short chain fatty acids acetic acid and butyric acid, increase ketone body concentrations. However, only one study has been performed in humans. This demonstrated that butyric acid is more ketogenic than either leucine or an 8-chain monoglyceride. Medium-chain triglycerides (MCTs) increase BOHB in a linear, dose-dependent manner, and promote both ketonaemia and ketogenesis. Exogenous ketones promote ketonaemia but may inhibit ketogenesis.
**Conclusions**. There is a clear ketogenic effect of supplemental MCTs; however, it is unclear whether they independently improve time to NK and reduce symptoms of keto-induction. There is limited research on the potential for other supplements to improve time to NK and reduce symptoms of keto-induction. Few studies have specifically evaluated symptoms and adverse effects of a ketogenic diet during the induction phase. Those that have typically were not designed to evaluate these variables as primary outcomes, and thus, more research is required to elucidate the role that supplementation might play in encouraging ketogenesis, improve time to NK, and reduce symptoms associated with keto-induction.

Corresponding author
Cliff J. d C. Harvey, cliff@hpn.ac.nz

## INTRODUCTION

Very-low-carbohydrate ketogenic diets (VLCKDs) are becoming increasingly popular for mainstream and athletic use for a range of outcomes including weight-loss and maintenance (*Bueno et al., 2013*), improved satiety and a reduction in hunger (*Paoli et al., 2015*; *McClernon et al., 2007*; *Johnstone et al., 2008*). The diet also offers specific benefits for health conditions ranging from neurological disorders, obesity, and diabetes and other conditions on the spectrum of metabolic syndrome, and offers potential for the adjunct treatment of various cancers (*Lefevre & Aronson, 2000*; *Keene, 2006*; *Levy et al., 2012*; *Henderson et al., 2006*; *Neal et al., 2008*; *Paoli et al., 2013*; *Sumithran & Proietto, 2008*; *Maalouf, Rho & Mattson, 2009*; *Castro et al., 2015*; *Varshneya et al., 2015*; *Kulak & Polotsky, 2013*). Ketogenic diets elicit a state of ketosis known as 'nutritional ketosis' (NK), a state of hyperketonaemia distinct from pathological ketosis such as diabetic ketoacidosis (DKA) (*Krebs, 1966*). Ketosis refers to the production of ketone bodies, derived from fats (and some amino acids) for use as an alternative fuel in times of fasting or drastic carbohydrate restriction. A restriction of carbohydrate, either by fasting or by restricting dietary carbohydrate, results in reduced insulin levels, thereby reducing lipogenesis (the creation of fats) and fat accumulation. When glycogen reserves become insufficient to supply the glucose necessary for normal β-oxidation of fat, via the provision of oxaloacetate in the Krebs cycle, acetyl-CoA is then used instead in the biosynthesis of ketone bodies via acetoacyl-CoA and β -hydroxy- β-methylglutaryl-CoA (*Lehninger, Cox & Nelson, 2008*) to ensure provision of fuel to the Central Nervous System (CNS), which usually relies on glucose. The process of ketogenesis further allows coenzymes to be freed to ensure continued fatty-acid β-oxidation (*Lehninger, Cox & Nelson, 2008*). To elicit this carbohydrate restriction, while also providing sufficient alternate fuel to ensure sustainability of the diet, i.e., in comparison to fasting to achieve ketosis, VLCKDs have been used to encourage ketosis. Early research on KDs focussed on children with epilepsy and for this purpose, a VLCKD typically consists of a 3:1 to 4:1 ratio of lipid to non-lipid. This treatment for epilepsy was pioneered at Johns Hopkins University Hospital (*Livingstone, 1972*; *Livingston, Pauli & Pruce, 1977*), and is referred to as a 'classic' or 'standard' ketogenic diet.

Ketogenic diets are now commonly applied, for a range of desired outcomes, and with differing definitions of what constitutes a ketogenic diet. Both low-energy diets and VLCKDs with fewer than 50 g of carbohydrate per day typically result in BOHB levels of $\geq 0.5$ mmol L$^{-1}$ (*Gibson et al., 2015*). This threshold has been used as a cut-off point for entry into ketosis by Guerci and colleagues (*Guerci et al., 2003*), and is commonly applied as a marker for entry into NK in the nutrition field, as compared to the typically higher levels expected in the medical field to elicit beneficial effects for seizure control in epileptic children (*Gilbert, Pyzik & Freeman, 2000*).

### Time to ketosis

There is a paucity of research that identifies specific time points to the now-common definition of NK, as defined by BOHB levels of $\geq 0.5$ mmol L$^{-1}$ (*Gibson et al., 2015*; *Guerci et al., 2003*). In a study comparing fasted ketogenic protocols to a more gradual initiation

of a ketogenic diet, Bergqvist and colleagues observed that participants fasting, achieved mean levels of $\geq 0.5$ mmol $L^{-1}$ BOHB, on the day following initiation of the diet, whereas those on a 1:1 ketogenic diet (by weight) achieved the same level two days after initiation of the diet (*Bergqvist et al., 2005*). Other studies have measured either tangentially or directly, the achievement of 'ketosis' but have not specifically identified the time at which a level of $\geq 0.5$ mmol $L^{-1}$ was achieved. Berry–Kravis and colleagues observed a mean time to ketosis (urinary >80 mg/dl) of 42 h (*Berry-Kravis et al., 2001*) Wirrell and colleagues have demonstrated a mean time to ketosis of 33 and 58 h for any trace of urinary ketones or 'good ketosis' (of >0.8 mmol $L^{-1}$) respectively (*Wirrell et al., 2002*). *Wusthoff et al. (2010)* recorded two cases of adults with prolonged nonconvulsive status epilepticus in which 'stable ketosis' was achieved after eight and 10 days respectively, 3.6 and >1.6 mmol $L^{-1}$, but the definition for ketosis, in this study, was not mentioned and we cannot extrapolate the time to NK as defined in clinical nutrition. *Strzelczyk et al. (2013)* suggested ketosis as the presence of urinary ketones, some 3.5 days after initiation of a ketogenic diet, but at that time participants had achieved serum BOHB of 3.6 mmol $L^{-1}$. Hoorn and colleagues observed no difference between fasted and non-fasted ketogenic protocols for time to ketosis, without specifically describing their definitions for ketosis or the time to ketosis itself (*Kang et al., 2007*; *Chul Kang et al., 2005*).

So, while the achievement of ketosis has been described in the medical literature, there are inconsistencies in the measurement of, and definition for ketosis in these papers.

## Adverse effects of keto-induction–the 'keto-flu'

Adaptation to a VLCKD, or 'keto-induction', and the achievement of NK, when transitioning from a standard, higher carbohydrate diet, can cause various unpleasant effects (*Hartman & Vining, 2007*). Symptoms of keto-induction are predominantly constipation, headache, halitosis, muscle cramps, diarrhoea, and general weakness and rash (*Yancy Jr et al., 2004*; *Kang et al., 2004*). These occur because of increased urinary sodium, potassium and water loss in response to lowered insulin levels (*Hamwi et al., 1967*; *De Fronzo, Goldberg & Agus, 1976*; *DeFronzo, 1981*; *Tiwari, Riazi & Ecelbarger, 2007*), greatest between days 1–4 of a fast or ketogenic diet (*Hamwi et al., 1967*), and transient reductions in glucose provision to the brain, observed to occur on days 1–3, with blood glucose normalising after day four (*Harber et al., 2005*). Constipation may result from reduced food volume or reduced fibre intake, although this finding could be due to the groups that have been studied, which have included children with disabilities, who commonly experience constipation due to immobility (*Kang et al., 2004*).

These symptoms are often referred to in the mainstream and grey literature as 'keto-flu' but are not well illustrated in the scientific literature. For example, a Google search returns over 22,000 results for the term "keto-flu," but the same term searched in MEDLINE Complete, CINAHL Complete, Alt HealthWatch, Food Science Source, SPORT Discus with Full Text, Psychology, and the EBSCO Behavioural Sciences Collection returns no results. Several studies have described adverse effects during ketogenic diets but to our knowledge, no studies have specifically described symptoms of keto-induction in the short time between commencing a ketogenic diet and the achievement of NK.

Adverse effects resulting from a VLCKD are likely to reduce compliance and tolerability (*Vining et al., 1998*), and thus affect the efficacy of these diets as clinical interventions.

There have been several methods suggested to reduce symptoms of keto-induction and to reduce the time taken to achieve NK, including the ketogenic amino acid leucine, short chain fatty acids, medium chain fatty acids, and exogenous ketones.

The aim of this paper, therefore, is to elucidate the evidence for and against commonly applied nutritional supplements, purported to be ketogenic, to inform clinical practice in the growing field of ketogenic diets for common-use. This paper reviews the available scientific literature relevant to improvements in time to ketosis and symptoms of keto-induction, resulting from these nutritional supplements.

## METHODS

PubMed, Science Direct, CINAHL, MEDLINE, Alt Health Watch, Food Science Source and EBSCO Psychology and Behavioural Sciences Collection electronic databases were searched online. Various purported ketogenic supplements, arising from a qualitative appraisal of forums, social media, message boards, and Google searches for ketogenic supplements, were searched along with the terms ''ketogenic diet'', ''ketogenic'', ''ketosis'' and ketonaemia (/ketonemia). Additionally, author names and reference lists were used for further search of the selected papers for related references. There is a paucity of studies on time to NK and mitigation of symptoms of keto-induction an as data related to the effects of various supplements on time to induction of ketosis and on symptoms of keto-induction are limited, and there is a lack of homogeneity between study objectives, outcomes, and measures, a narrative review style was chosen.

## RESULTS

### Leucine

Leucine and lysine are solely ketogenic amino acids. Thus, they do not contribute to gluconeogenesis. Higher leucine (and isoleucine) concentrations result from a ketogenic diet and are related to reduced glutamate-to-GABA ratio and this might explain some of the anti-seizure activity of a ketogenic diet in epilepsy (*Roy et al., 2015*). There appears to be a high affinity of kidney cells for ketogenesis from leucine (*Noda & Ichihara, 1976*).

Progression of fasting increases the conversion of leucine to ketone bodies and peripheral tissue is catabolised to provide leucine for ketogenesis (*Kulaylat et al., 1988*). Leucine can also be degraded in rat astroglial cells to the ketone bodies, including BOHB, and when released by these cells, used by neighbouring neurones as a fuel substrate (*Bixel & Hamprecht, 1995*). Leucine also results in hepatic ketogenesis (*Holecek et al., 2003*). Studies in mice have shown that while ingested L-leucine can reduce seizure activity similarly to a KD, it does not independently increase blood levels of BOHB (*Hartman et al., 2015*). Evangeliou and colleagues have demonstrated that the addition of 20 g per day of BCAAs, including 9 g of leucine, in 17 children with intractable epilepsy, altering the ratio of lipid to protein from 4:1 to around 2.5:, had no effect on ketosis, along with greater reductions

in seizure activity. The authors postulated that this could be due to the ketogenic effect of leucine, but may also result from a greater availability of BCAAs (*Evangeliou et al., 2009*).

## Short chain fatty acids

Short-chain fatty acids (SCFAs) have carbon chains between two and five in length. These fatty acids include acetic acid (C:2), propionic acid (C:3), butyric acid (C:4), and valeric acid (C:5). Short chain fatty acids, especially butyric acid, are used extensively as a fuel substrate by intestinal epithelial cells (*Wong et al., 2006*). It is generally accepted that chain length affects the relative deposition of fatty acids into either lymph or the portal vein (*Mu & Høy, 2004*). Those short-chain fatty acids that escape metabolism by epithelial cells are, therefore, primarily absorbed via the hepatic portal vein and do not require 'bundling' with micelles and chylomicrons for absorption (*Kuksis, 2000*). The highest quantities of short-chain fatty acids have been observed in portal blood, followed by hepatic, and far less in peripheral blood (*Cummings et al., 1987*). Thus, they bypass the usual route of absorption (for the more common long-chain fatty acids) into the lymphatics and deposition into the bloodstream via the subclavian vein, and instead, are transported via the hepatic portal vein to the liver where they can be converted into the ketone bodies (*Bugaut, 1987*; *Bourassa et al., 2016*; *Stilling et al., 2016*).

## Acetic acid

Acetic acid is a two-carbon SCFA. It comprises approximately 4–20% of vinegar. Vinegar has been demonstrated to improve postprandial insulin sensitivity in healthy and diabetic people and improve glycaemic responses to meals (*Johnston, Kim & Buller, 2004*; *Liljeberg & Björck, 1998*; *Brighenti et al., 1995*). Urinary excretion of acetone (a ketone body) is increased in phloridzinised dogs and fasting rats after feeding with acetic acid (*MacKay et al., 1940*). Acetone is the spontaneous breakdown product of the ketone bodies acetoacetate and BOHB. Thus, it is likely that acetic acid is ketogenic, and has additional benefits for overall metabolic health, however, no research has been performed on acetic acid and its specific effects on the induction of ketosis or mitigation of keto-induction symptoms in humans. Interestingly, vinegar is commonly prescribed as a 'free food' in ketogenic diet trials (*Rother, 2007*; *Perez-Guisado & Munoz-Serrano, 2011*; *Nebeling & Lerner, 1995*), and may provide an under-recognized stimulus for ketogenesis.

## Butyric acid

Butyric acid (BTA) is a four-carbon, short-chain fatty acid found in the milk of ruminants and present in small amounts in many dairy foods. Most BTA in humans is produced by microbial intestinal fermentation of dietary fibre and resistant starch. Most of the butyric acid produced by this fermentation of starches is absorbed and used directly by colonocytes, with most of the remainder absorbed into the hepatic portal vein, and transported to the liver where it can be converted to ketone bodies (*Bourassa et al., 2016*; *Stilling et al., 2016*). A small amount is absorbed directly from the large colon and enters systemic circulation, to be used directly by peripheral tissue (*Bourassa et al., 2016*). Butyrate exerts effects directly on the colonic mucosa, including inhibition of inflammation and carcinogenesis, decreasing oxidative stress, and promotion of satiety (*Hamer et al., 2008*; *Fung et al., 2012*).

Thus, it serves an important role in preserving the health of the colon, microbiota, and may have other beneficial roles for general and systemic health. Animal studies on the ketogenic potential of butyrate are mixed. For example, silage butyrate content has been shown to provide no significant effect on subclinical ketosis in dairy cows (*Samiei et al., 2015*), however, sub-clinical ketosis is higher in those receiving silage higher in butyrate content (*Vicente et al., 2014*).

In a recent study in humans, the effect of L-leucine, octanoyl-monoacylglycerol (O-MAG), a monoglyceride consisting of an 8-carbon fatty acid, L-carnitine, and butyric acid on acetoacetate and BOHB were studied. Both 2 g and 4 g of butyric acid were demonstrated to be more ketogenic than either 5 g of leucine, or 5 or 10 g of O-MAG (*St-Pierre et al., 2017*).

## Medium chain triglycerides

In medium chain triglycerides (MCTs) two-to-three of the fatty acid chains attached to the glycerol backbone are medium in length. These medium-chain fatty acids (MCFAs) are comprised of a 6–12 carbon chain. The MCTs are: caproic (C6), caprylic (C8), capric (C10) and lauric acid (C12) (*Marten, Pfeuffer & Schrezenmeir, 2006*). Similar to the short-chain fatty acids and unlike long-chain triglycerides (LCTs), MCTs do not require the actions of bile, nor micellar-chylomicron mediated absorption into the lymphatics and instead are diffused directly into the hepatic portal vein and preferentially converted into bio-available ketone bodies in the liver. Huttenlocher and colleagues first demonstrated that diets containing fewer calories from lipids than a 'classic' ketogenic diet—around 60%–75% of calories—can induce NK if they include a high proportion of medium chain triglycerides (MCTs) (*Huttenlocher, Wilbourn & Signore, 1971*). A VLCKD with 60% of energy derived from MCTs, a three-fold greater intake of carbohydrate (18% vs. 6%) and a ∼50% (7% vs. 10%) increase in protein compared to a standard ketogenic diet induces NK with no appreciable difference in BOHB levels (*Huttenlocher, 1976*).

Dietary MCTs are also known to promote both ketonaemia and ketogenesis in animals (*Bach et al., 1977*; *Yeh & Zee, 1976*) and humans with and without health conditions (*St-Onge et al., 2003*; *Yajnik et al., 1997*). MCTs promote ketonaemia and ketogenesis (useful to reduce the risk of night-time hypoglycaemic coma) in those with carnitine palmitoyltransferase deficiency, a rare genetic condition which inhibits the ability to produce ketone bodies from long-chain fatty acids (*Bonnefont et al., 1989*; *Bougnères et al., 1981*). MCTs also increase BOHB when calorically dose-matched to either LCTs or carbohydrate in single feeding and non-ketogenic diet studies (*Decombaz et al., 1983*; *Seaton et al., 1986*; *Yost & Eckel, 1989*; *Krotkiewski, 2001*). When fed intravenously, MCTs increase ketogenesis when compared to both structurally similar fats (*Mingrone et al., 1993*) and LCTs (*Jiang et al., 1993*; *Lai & Chen, 2000*). However, ketogenesis is reduced by the simultaneous application of glucose (*Kolb & Sailer, 1984*). It has been demonstrated by Sandstrom and colleagues that in a hypercaloric diet, there are increased BOHB levels observed with the application of MCTs that aren't seen in a hypocaloric state (*Sandström et al., 1995*).

MCTs increase BOHB in a linear and dose-dependent fashion. For example, when eleven pre-term infants were fed formulas with either 25% or 50% of fat calories coming from

MCTs for at least 96 h (30 kcal/ml, around 50% calories from fat in total, 10% protein, 40% carbohydrate) the 50% MCT formula resulted in a mean plasma level of BOHB of $0.14 \pm 0.03$ mmol/L/, a nearly three-fold increase over the lower MCT formula ($0.06 \pm 0.01$) (*Wu et al., 1986*).

While there is a paucity of research on the effect of MCTs on the time taken to achieve NK, MCTs are demonstrably ketogenic and thus, allow induction of NK with lower proportions of fat in the diet, than that used in 'classic' 3 or 4:1 lipid to non-lipid (or 'ketogenic ratio') protocols. When 'classic' ketogenic diets with a greater than 3:1 ratio of lipid to non-lipid are compared to MCT ketogenic diets with 60% of calories from MCT, NK can be achieved with a lower lipid intake. Huttenlocher first observed higher BOHB levels in children with epilepsy aged 2–9 years, at up to one month on an MCT ketogenic diet, and marginally lower after this time, when compared to a classic ketogenic diet, although these differences were not significant (*Huttenlocher, 1976*). In a study of 55 children with severe epilepsy, Schwartz and colleagues found modified ketogenic diets, MCT ketogenic diets, and classic ketogenic diets to all be 'ketogenic' (inducing NK) with peak ketone body concentrations of approximately 1 mmol/L, 1.5 mmol/L and 4 mmol/L respectively, after three weeks on the differing ketogenic protocols (*Schwartz, Boyes & Aynsley-Green, 1989*). Nine children were subsequently trialled on a second diet and profiled three weeks later. Cumulative results over 24 h of metabolic testing demonstrate that expression of ketone bodies rises (in order) from a normal diet (little change) to a modified MCT diet, an MCT ketogenic diet, and the greatest rise in ketone bodies over 24 h resulting from a classic (4:1) ketogenic diet. In a 12-month study, a classic ketogenic diet resulted in higher levels of BOHB (and acetoacetate) over all time periods (three, six, and 12 months) but this was only statistically significant at three and six months ($p < 0.001$) (*Neal et al., 2009*).

After ingestion of MCT at a dosage of 30 g MCT/m (*Paoli et al., 2015*) body surface area by nine children (in a study of seizure control), BOHB levels rose progressively after administration from a mean of $0.2 \pm 0.1$ mmol/L after an overnight fast to $1.05 \pm 0.3$ mmol/L at 180 min. Participants reached NK on average at 30–60 min with most participants in NK by the 90th minute, but there was significant variation in BOHB between individuals (*Ross et al., 1985*). With a lower dosage of 7.5 g of MCT taken three times per day after an acclimation period of 5 g MCT taken three times per day for one week, plasma BOHB was higher, yet not inducing NK (*Courchesne-Loyer et al., 2013*).

## Exogenous ketones

Exogenous ketone supplements provide BOHB directly to the body without requiring ketogenesis and without concurrent elevations in free fatty acids (*Veech, 2014*). They are considered to be a safe and effective way to increase ketone body concentrations (*Hashim & Van Itallie, 2014*). Ketone supplements demonstrate promise as potential adjunct treatments for brain injury (*White & Venkatesh, 2011*), cancer (*Poff et al., 2015*; *Poff et al., 2014*), Angelman syndrome (*Ciarlone et al., 2016*), for reducing inflammation by suppressing activation of the NLRP3 inflammasome (*Youm et al., 2015*), and Alzheimer's disease (*Kashiwaya et al., 2013*). Ketone supplements might also improve fueling during exercise, reduce lactate production, and improve performance due to glucose sparing

(*Okuda et al., 1991*), and have positive effects on anxiety (*Kashiwaya et al., 2013*), and mental performance and memory (*Kashiwaya et al., 2013*).

Exogenous ketone supplements are available as either salts or esters of BOHB. Supplements containing ketone salts (KS) are some combination of sodium-, magnesium-, calcium or potassium-BOHB, and are available commercially from several companies under patent (*D'Agostino, Arnold & Kesl, 2015*). Ketone esters (KEs) at the time of writing, are only available for research, primarily as 1,3-butanediol monoester of BOHB (*Hashim & Van Itallie, 2014*) and thus, the animal and human research has mostly focused on the use of ketone esters. Both ketone esters and salts elevate BOHB to levels consistent with NK (*Holdsworth, Cox & Clarke, 2016*), with ketone esters having greater effects on ketonaemia with ketone salts providing significantly higher reporting of gastrointestinal symptoms (*Stubbs et al., 2016*). Ketone salts might provide a greater potential for long-term side effects if the inorganic ion load delivered is excessive for the individual (*Stubbs et al., 2016*). Conversely, R-1,3-butanediol from ketone monoesters is readily metabolized in the liver to AcAc (*Clarke et al., 2012*). *Clarke et al. (2012)* detected no R-1,3-butanediol in the plasma of participants taking a ketone monoester supplement, except at the highest dosage of 714 mg/kg body weight, at which dose plasma R-1,3-butanediol was detectable at a level of $\leq 1.0$ mmol/L and was undetectable 4 h later.

At a dosage of 395 mg/kg bodyweight, KE increased BOHB in healthy volunteers from 0.2 mmol/L ($\pm 0.02$) at baseline to 3.3 mmol/L ($\pm 0.2$) one hour later (*Stubbs et al., 2015a*), and from 0.16 mmol/L ($\pm 0.02$) at baseline to 3.16 mmol/L ($\pm 0.14$) (*Stubbs et al., 2015b*) The same dose has been used to determine the effect on ketonaemia of KE taken with or without a meal. BOHB concentration (one-hour post-KE) was lower in those having taken a meal, but both groups achieved levels of ketonaemia consistent with NK; 2.1 mmol/L ($\pm 0.2$) and 3.1 mmol/L ($\pm 0.1$) respectively (*Stubbs et al., 2015c*). In a study using higher dosages (0.573 g/kg BW) in healthy male athletes performing an hour of bicycle exercise at 75% of maximal exercise intensity BOHB levels rose from 0.1 to 3.4 mmol/L ($p < 0.01$) following ketone drinks (*Cox et al., 2015*).

While it is clear that exogenous ketones increase serum BOHB, they are not ketogenic, and may, in fact, inhibit endogenous ketone production (*Balasse & Neef, 1975*). In other words, they promote ketonaemia but do not encourage the creation of ketone bodies in the liver. So, it is more accurate to say that exogenous ketones mimic the effects, many of which are positive, of NK, rather than inducing it.

## CONCLUSIONS

It's unclear at this time whether an elevation in ketones over and above NK would mitigate the effects of keto-induction. It has, for example, been observed that mood is improved within the first two weeks of a diet irrespective of macronutrient composition (*Rosen et al., 1985*), and only one study, to our knowledge, has demonstrated a correlation between ketone levels and memory performance (*Krikorian et al., 2012*).

Except for MCTs, there is limited research on the ketogenic potential of nutritional supplements, especially in human subjects. While the ketogenic amino acid leucine may

not independently encourage ketogenesis to levels consistent with NK, more research is required, and the effect on time to NK and symptoms of keto-induction, particularly in a classic KD, are at this stage unknown.

Similarly, there is a paucity of research on the short-chain fatty acids and their effects on ketogenesis. Their mode of absorption and metabolism, like that of MCTs, but perhaps even more rapid, hints at a potential role for encouraging ketogenesis, and thus, the potential for improving time to NK and reducing symptoms of keto-induction.

There is a considerable amount of research demonstrating that MCTs promote both primary ketonaemia resulting from the conversion of medium chain fatty acids liberated from MCTs into bio-available ketone bodies, and longer-term ketogenesis by facilitating keto-adaptation. Expression of the ketone body BOHB is increased in a linear, dose-dependent manner in response to oral loads of MCT but it is unclear whether MCTs independently improve time to NK. Modified MCT ketogenic diets do not significantly hasten the induction of NK over a classic ketogenic diet with a minimum of three parts lipid to one part non-lipid, but they do allow NK to occur in diets containing greater amounts of non-lipid macronutrients.

There has, however, been little research performed on the application of MCTs to classic ketogenic diets and whether, if applied, they would; (a) improve time to NK, (b) result in significantly higher levels of BOHB, and (c) significantly reduce symptoms of keto-induction. It is also unknown if, in the context of a ketogenic diet, MCTs provide additional benefits, for example for physical and mental performance and mood.

Exogenous ketones are unlikely to be ketogenic per se, and may inhibit ketogenesis, however, the rapid and substantial elevation of BOHB offers potential to mitigate effects of keto-induction, and thus, could play a role in improving adherence to a ketogenic diet. Newport et al. have reported improvements in mood and cognitive performance resulting from ketone ester treatment over 20-months in an Alzheimer's Disease case. In this case, cognitive performance tracked plasma BOHB concentrations. In a direct, dose-matched comparison, Kesl and colleagues evaluated the effects of ketone esters, salts, MCTs, and MCT + KS on blood BOHB in Sprague-Dawley rats at a dose of 5 g/kg. At 0, 30, and 60 min and 4, 8, and 12 hrs post administration (by intragastric gavage) KS + MCT and MCT supplementation rapidly elevated and sustained significant BOHB elevation compared to control for the duration of the 4-week study. Ketone salts did not significantly elevate BOHB at any time point tested compared to controls. Ketone ester supplements significantly elevated BOHB levels for the duration of the 4-week study. This further demonstrates, albeit, in non-human subjects, the superiority of KE to KS for elevating BOHB, and the utility of MCT for the same purpose, but is likely to limited in applicability to health and performance as we have seen demonstrable increases in BOHB, consistent with NK levels with supplementation of KS in humans (*Holdsworth, Cox & Clarke, 2016*; *Stubbs et al., 2016*). Research performed on exogenous ketone supplements is, at this time, highly preliminary, and has been predominantly performed using animal subjects. Further clinical research is required to translate the potential benefits seen in these studies, to human models of disease and disorder.

This review was limited by a dearth of studies demonstrating the effect of supplementation on the time taken to achieve ketosis as defined by the *lingua franca* of NK, $\geq 0.5$ mmol L $-1$ and on symptoms of keto-induction during this time.

While studies have described symptoms arising from a ketogenic diet, few studies have specifically evaluated symptoms and adverse effects of a ketogenic diet during the induction phase, and the studies that have been performed typically have not been designed to evaluate these as primary outcomes, and thus, our conclusions are extrapolated from a variety of sources. There is also little consensus on whether greater levels of BOHB (over and above NK threshold) are, in fact, associated with fewer symptoms of 'keto-flu', nor for that matter with improved outcomes but as previously noted, Newport and colleagues have observed a linear correlation between mood and cognition, and BOHB levels (*Newport et al., 2015*). Adverse effects associated with the induction of NK might cause increased drop-out rates and preclude some of the positive effects for those that would otherwise benefit from a VLCKD. For example, Yancy and colleagues noted an 8% overall dropout rate due to difficulties adhering to an LCHF diet, with a further 5% withdrawing from their study due to adverse effects (*Yancy Jr et al., 2004*). High attrition rates due to tolerability and gastrointestinal side effects have also been noted in childhood epilepsy research utilising VLCKDs (*Levy et al., 2012*; *Chul Kang et al., 2005*).

Preliminary research suggests that increased BOHB levels and a faster time-to-NK might improve the acceptability of the KD and improve compliance rates, but more research is required to understand the role that supplementation could play in encouraging ketogenesis, improving time to NK, reducing symptoms associated with keto-induction, and the effect this might have on improving adherence to, and outcomes from a VLCKD.

## ACKNOWLEDGEMENTS

We acknowledge the support of our colleagues at the Human Potential Centre, AUT University, especially Eric Helms and Simon Thornley, who helped with the final editing of this document, and Darrell Bonetti who provided guidance on the direction of the review.

### Funding

All funding for this work was provided by the Auckland University of Technology. The funders had no role in study design, data collection and analysis, decision to publish, or preparation of the manuscript.

### Grant Disclosures

The following grant information was disclosed by the authors:
Auckland University of Technology.

### Competing Interests

The authors declare there are no competing interests.

## Author Contributions

- Cliff J. d C. Harvey conceived and designed the experiments, performed the experiments, analyzed the data, contributed reagents/materials/analysis tools, prepared figures and/or tables, authored or reviewed drafts of the paper, approved the final draft.
- Grant M. Schofield and Micalla Williden conceived and designed the experiments, contributed reagents/materials/analysis tools, prepared figures and/or tables, authored or reviewed drafts of the paper, approved the final draft.

## Data Availability

The following information was supplied regarding data availability.

As a narrative review, there is no data set associated with this paper.

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
