# Peer review of "The use of nutritional supplements to induce ketosis and reduce symptoms associated with keto-induction: a narrative review"

_PeerJ, doi:10.7717/peerj.4488_

## Round 0.1 · original submission · Major Revisions

Please carefully address the comments of the reviewers.

·

Basic reporting

The authors aimed to review the current literature available on nutritional ketosis, the use of ketone supplements and how they can be used to mitigate negative symptoms associated with keto-induction. Considering the recent expansion in commercially available products aimed at facilitating the adherence to KD, it is crucial to better understand the actual scientific evidence supporting the claims currently made.

The manuscript is well structured and well-written. However, a few edits would improve the quality of this report:

1. References:
1.1. No reference has been listed to substantiate any of the sentences written in the ‘Short chain fatty acids’ section (Lines 92-98)
1.2. Lines 111-114 also need reference(s)
1.3. St. Pierre et al. mentioned in Line 122 is not listed in the References section

2. The exogenous ketones are simply described as either salts or esters of BOHB. Since describing the use of ketone supplements to facilitate nutritional ketosis is a key objective of this review, it would be helpful to have a more detailed description of these supplements (types of salts, types of esters, biodegradation and bioavailability, adverse effects, proposed applications etc.)

3. Additional and relevant data comparing ketone salts and MCT (described in Kesl et al., Nutrition & Metabolism, 2016; 13:9) is available and can be included to further improve this narrative review on supplements used to induce ketosis

4. It is relevant to address that there’s lack of consensus on whether achieving greater ketosis necessarily means better physiological outcomes. Some of the questions in the field that are of importance and could be expanded upon in the text:
4.1. Would achieving higher levels of ketosis (by supplementing with exogenous esters, salts or MCTs) shorten the keto-induction period and/or enhance the physiological benefits observed with nutritional ketosis?
4.2. Is there any available scientific evidence that ketone supplements do actually mitigate the ‘keto-flu’ symptoms associated with the keto-induction period?
4.3. Does chronic supplementation with exogenous ketones impair endogenous ketogenesis (as observed in Balasse et al., 1975)?

5. When discussing MCTs, the authors describe their ketogenic effects, suggesting that acute ingestion of MCTs induces nutritional ketosis. However, it is important to differentiate between hepatic production of ketone bodies due to acute ingestion of MCTs and the biochemical shift in fuel utilization that occurs during prolonged nutritional ketosis (the physiological process described as keto-induction or keto-adaptation). I suggest addressing this difference and possibly emphasizing the acute/transient effects elicited by single or acute consumption of exogenous ketones, resulting in peripheral ketosis but not necessarily keto-adaptation. On that note (and according to lines 159-161), would the MCT ketogenic diet likely shorten the keto-induction period when compared to the classic KD?

6. The language is appropriate but a few typos were found:
6.1. Abstract: Line 20: dose-dependant instead of dose-dependent
6.2. Introduction:
6.2.1. Line 46: references 22 and 23 are listed not separated by a comma
6.2.2. Line 50: Fibre instead of fiber

Experimental design

A summary table with relevant articles on ketogenic diets (classic, modified MCT diet, MCT ketogenic diet and supplements) listing preclinical and clinical studies and main observations (levels of ketosis, time to achieve ketosis, physiological outcomes, adverse effects etc.) would provide greater support and structure for this narrative review.

Validity of the findings

7. The topic of ketogenic diets and ketone supplements has risen in popularity due to well-known benefits in weight management, diabetes, athletic performance and metabolic health. However, it is important to emphasize that, although this approach is promising as potential therapeutic intervention for many other conditions, clinical data is still lacking. For instance, the positive effects in anxiety described by Kashiwaya et al., (2013) were observed in transgenic models of amyloid and tau deposition. An additional report of ketone supplementation reducing anxiety-related behavior in rodents was published last year (Ari et al., Front Mol Neurosci, 2016; 9:137) using two different rat strains. Clinical studies on ketogenic diets and supplementation in populations of mental disorders patients are still lacking. Studies based on animal models of mental disorders have limited generalizability when translated into the same conditions in humans. Caution should be exercised and clinical studies remain necessary.

8. The same caution can be extrapolated to other areas such as cancer, autism spectrum disorders, Alzheimer’s disease etc., where mostly case reports are currently published and the vast majority of data is originated from animal studies and anecdotal reports. This does not invalidate these approaches, as sound medical hypothesis have been published (as in the cases of metabolic dysfunction associated with cancer and Alzheimer’s disease) but it does highlight the need for well-designed randomized, clinical trials to support the available hypothesis and preclinical data.

Additional comments

Given the abundance of anecdotal reports on the health applications of ketogenic approaches, this review is of particular interest and value to the scientific community. I thank the authors for addressing such an important topic, compiling evidence for such a promising metabolic therapy implicated in diverse health conditions.

Reviewer 2 ·

Basic reporting

I believe there is insufficient background and context to the subject provided; the diet used for mainstream and athletic use is generally a very different diet from the one used for neurological disorders such as epilepsy. In order to write this paper - these 2 separate diets need to be explained; the rest of the article then should follow suit: which diet along the course of this paper are the authors referring to?
IF it is the medically managed KD then I disagree with many statements that have been made: For example: there is a wealth of literature that reports the length of time to "nutritional ketosis" and the side effects that relate to this - none of which have been listed by the authors. This is important as it is the question of the study. What Is meant by an "improvement in time to ketosis?" line 65? Is a short time an improvement? Without a parameter for time to ketosis I fail to understand how you could study improvements in time to ketosis!

If it is a low carbohydrate diet then the parameters for carbohydrate and fat need to be provided.
The mechanism for ketone production by the Classic KD is not explained; it is the combination of high fat AND low carbohydrate - causing a shift from burning glucose as first from of fuel to fat; this shift in metabolism is what leads to all of the short and long term side effects. Many statements written in the paper appear to have negated this basic premise.
Many words are not considered scientifically "sound" in keto world - words such as "keto-flu" "nutritional ketosis" and Very low CHO KD are not considered scientifically credible.

Experimental design

Research question is not well defined; see basic reporting above; more background needs to be provided in order to understand the study aims better.
Would suggest to be considered scientifically more credible EMBASE should have been searched.
I struggle to comment on the manuscript regarding leucine; it may be a Ketogenic AA but needs to be given in the context of a ketogenic environment to work? There is no need for it to act as such unless stores are depleted and it is being used as an alternative source of fuel. Vinegar - interesting! BUT the only reason it is prescribed as a "free food" in KD is it is carbohydrate free; from a practical point of view; so to are dried herbs and spices, salt and pepper and lettuce!
MCTs are useful in the space of KD - the reason they are more Keto yielding has been stated incorrectly - it is merely because for some metabolisms the capacity of B oxidation has been exhausted and so another form of fat; metabolised via a different pathway can effectively produce ketones. MCT used for CPT II and II is used as the oxidation of LCT is not possible and those patients require an alterative source of fuel to avoid metabolic decompensation.
Exogenous ketones - their role has been established; this is not clearly defined. D'Agostino is the expert and should be considered extensively here; unfortunately only mice studies have used those EK used in research - these appear to work but are unavailable to the public and taste awful!

Validity of the findings

Believe this study could be interesting but basic parameters need to be addressed in [particular what diet is being referred to and is supplementation on top of a KD or on top of a "normal diet".
Authors need to examine literature more thoroughly to establish the length to ketosis ) it is defined in the medically managed diet; as to are the side effects of this stage.

Additional comments

Agree the authors have chosen a topical subject: many people are seeking a "quick fix" to feel better; lose weight, feel fuller etc. a supplement that could do that would be great. This paper needs more work to establish the basics and the context before discussing the potential for nutritional supplements.

---

## Round 0.2 · Minor Revisions

Please revise the manuscript according to the reviewer 2's comments.

·

Basic reporting

No comment

Experimental design

No comment

Validity of the findings

No comment

Additional comments

I thank you the authors for addressing the comments made by the reviewers. The manuscript has been significantly improved by the changes in the revised version submitted.

Reviewer 2 ·

Basic reporting

The language used is ambiguous and is not the vocabularly used in this field. The terms "nutritional ketosis" "keto-flu" "Very Low carbohdyrate ketogenic diet (incorrect!) " "natriuresis, kaliuresis" are rarely if ever used amongst health care professionals looking after patients on the medically managed Classical ketogenic diet. This is the diet that I am believe the authors refer to as stated: line 48 - "consists of a 3:1 to 4:1 ratio". The article starts with VLCKDs are becoming increasingly popular for mainstream athletic use.....Line 31/32 but never refers to these group again. The literature cited has been taken largely from studies based on the KD for epilepsy and has stated that "there is a paucity of research available that identifies specific time points to NK" - line 73. In the epilepsy community functional "ketosis" ie: that which has the potential for a beneficial effect on seizures and be considered a point at which the body has truly started to use fat as the first form of fuel is determined as > 2.0mmol/L not 0.5mmol/L; as stated by the authors - line 75. Again, amongst this community - it is well known that a 3:1 to 4:1 ratio diet followed strictly for 2-4 days will achieve levels 2-4mmol/L - which makes me question what diet you are reviewing?
I do not believe the article is structured well and does not report on key studies regarding Ketosis and time taken to achieve. The evidence for ketosis can be witnessed in many studies on dietary efficacy in the epilepsy community.

Experimental design

To review the literature the article uses various search engines. It is unclear how the words that were put into the search engines were linked to the "ketogenic supplements".. A ketogenic diet search alone will retrieve 100s of articles. No discussion is made of this data and reasons for excluding a huge body of literature.
Much of the literature that has been reported is either inconsequential or irrelevant to this discussion. This is evidenced by line 74 - Bergqvists paper - it states that participants fasting achieved mean levels of >0.5mmol/L BOHB the day after......the conclusions you have drawn suggest a lack of understanding in the clinical delivery of the diet. In addition, this paper is again children with epilepsy. The aim in these chidlren is > 2.0mmol/L. A level of 0.5mmol/L can be achieved in most of us by fasting overnight and having a delayed breakfast! Line 80 - Wirrell and colleagues, what ratio diet were they using? It would be more common to report in the literaure on those who do not achieve ketosis on a 3:1 - 4;:1 ratio quickly than those who do!

Validity of the findings

These findings may be true but I fail to see relevance in the article; For who is it to be used for? Children and adults with epilepsy who we know we can achieve ketone levels > 2mmol/L over a number of days, or athletes and the general public?

In addition, time to the ketosis level in children with epilepsy has little impact on tolerability of the diet. The difficulty in managing a diet which is providing 90% fat is one of the main reasons for dietary disbandoment. This is well documented also.

Additional comments

The article sets out to review scientifc knowledge on effects of methods listed on "time to NK" and on symptoms during keto - induction phase.
It may have achieved the goal of identifying MCT as ketogenic yielding (nothing new in the literature here) and excluded others but I fail to see the value of this research to anyone.

---

## Round 0.3 · Major Revisions

Please address the reviewer's concerns and suggestions.

Reviewer 2 ·

Basic reporting

Several improvements to this article have been made but key issues remain

1) Language often used contraindicated to topic and not used in scholarly articles; Very Low Carbohdyrate Ketogenic Diet is ambiguous; "keto-flu" - why discussed; why not symptoms as per aim of the study?
2) Existing literature in the area is missing and that employed often irrelevant or not the most appropriate (ie see lines 81-95 - these studies are taking groups of patients that are not comparable, hence no conclusion understandably drawn)

3) Improved structure required - very hard to follow - suggest: What is the aim of the study; why are you studying it? ie what is your experience with length of time to ketosis and what is considered an improvement? Currently without a defined length of time to ketosis how can you say what is an improvement? In addition, How debilitating are the side effects of keto induction (this needs defining as gives meaning to the paper) and would decreasing length of time really help this (practical experiecnce at an interantional level in the area of KD in epilepsy has shown not). Why were leucine etc chosen as supplements? ie number of articles from literature search please. For each supplement chosen address the hypotheses: link to length of time to keto induction and side effects. This will give your paper meaning.
Conclude with either use, consider using and why....

4) Results may be relevant to hypotheses but there no statements linking the two. An extensive literature search was conducted - what were the results of this? How many articles and in what journals highlighted leucine, SCFAs etc as relevant to time to NK? Symptoms during Keto induction phase? In addition; at conclusion of these sections - ie line 126-127; discusses leucine reducing seizure activity and not independently increasing blood levels of BOHB; neither statements linked to hypotheses and do so wonder why leucine was chosen?

Experimental design

Research question not well defined; Remain unsure as to the target group this work is trying to assist, in addition not clear that reserach fills any knowledge gaps.

Rigourous investigation not performed; studies have been reviewed without meaningful consistent; comparisons made. understanding of practical implication of diet appears to lack

Methods not replicable - as per results - see point 4) basic reporting

Validity of the findings

Replication of results would not be possible as per point 4) basic reporting
Unable to conclude any point to article: are any supplements worth or not worth considering as part of dietary management for the athlete? The child with epilepsy?
Could not say with conviction to use or not to use in either of these target groups after readig paper.

Conclusions from results sections do not link to oringial hypotheses; which itself is not well established.

Line 103 states "paper reviews the available scientific literature relevant to improvements in time to ketosis......" but nowhere is it defined what the time to ketosis is (as stated by the aurthor) and to what degree this is an issue.

Many changes have been made to protocols for ketogenic diet intiation in the patient with epilepsy over the past 10 years; studies report extensively on the success of switching from fasting to non-fasting initiation and the move from inpatient to outpatient initiation. Both of these changes were made to reduce symptoms of ketogenic diet initiation, occasionally at the expense of increasing time to ketosis - these protocols are tailored to the individual based on their preferences BUT generally most patients (with epilepsy) are able to achieve high ketosis (>2mmol/l) chronically on a KD protocol which has been initiated over a 2-5 day period; very few with any side effects at all. This work is all published. I would very much like to know how your work links in with this vast amount of literature, some at the highest level of reporting.

Additional comments

See glimmers of success with the publishing of this article but structure and improved linking statements are required to ensure its' success.

---

## Round 0.4 · accepted · Accept

The paper is now suitable for publication, congratulations.